# Strategies for Attention to Diversity: Perceptions of Secondary School Teaching Staff

**DOI:** 10.3390/ijerph17113840

**Published:** 2020-05-28

**Authors:** Rosa Goig Martínez, Isabel Martínez Sánchez, Daniel González González, José Luis García Llamas

**Affiliations:** 1Department of Research Methods and Diagnosis in Education I (MIDE I), Faculty of Education, Spanish University for Distance Education (UNED), 28040 Madrid, Spain; imsanchez@edu.uned.es (I.M.S.); jlgarcia@edu.uned.es (J.L.G.L.); 2Department of Research Methods and Diagnosis in Education (MIDE), Faculty of Education, Universidad de Granada, 18071 Granada, Spain; danielg@ugr.es

**Keywords:** attention to diversity, educational quality, quality management system, inclusive school

## Abstract

(1) Background: Attention to diversity constitutes an aspect that influences system quality and offers a perspective of the capacity of educational centres to respond to educational needs. The present study carried out an examination of the perceptions held by secondary school teachers and the level of importance conferred by them to the variables that should be integrated into plans and will influence the degree of compliance. (2) Methods: Quantitative descriptive research was performed using a survey to collect data from teachers at schools that had a Quality Management System available. (3) Results: Interaction with families is necessary to agree upon the centre’s objectives to address diversity and to define an optimisation strategy for resources in virtue of their availability within the centre. It is key to establish an appropriate teacher–student ratio to encourage compliance. (4) Conclusions: Teachers are the great pillars of quality education. Their perceptions are the route through which deficient aspects and the dimensions that must be improved when formulating these strategies can be recognised with attention to diversity.

## 1. Introduction

This research focuses on strategies for attention to diversity, carrying out an examination of perceptions and the level of importance conferred by teachers to the different variables that should be integrated within any plan. In line with this field of research, concepts emerge such as education quality, which directly links attention with diversity, bearing in mind that the educational system has to provide the real conditions in order to favour an educational experience adapted to the idiosyncrasies of its pupils [1,2].

Silverman [3] makes it clear that investigating the perceptions of teachers in relation to diversity is key for understanding the different approaches needed to be put in place depending on the specific educational context. Teachers work on different types of explicit knowledge that gather the knowledge of what to do, how to do it and why to do it. The second (tacit knowledge), is based on internalising through experience and/or observation of the facts. This knowledge is not shared or communicated. Thoughts, intuitions, expectations and perceptions are part of this type of cognition [4]. As a criterion to follow in this research, perceptions can be seen as a conceptual substratum that plays an important role in thought and action, providing points of view of the world and as concept organisers [5,6]. It is essential to consider perceptions as a set of positions that a teacher has about his or her practice in correspondence with issues related to teaching and learning [7]. In this sense, we focus it on the position that teachers have on the different variables that make up the plans of attention to diversity, for the achievement of a quality and inclusive education.

In this context, according to Araque and Barrio [8], attention to diversity is embedded within the framework for inclusive education and establishes as its starting point the recognition that multiple requirements exist in relation to the educational process each student undergoes. These requirements must be met in full by the educational system in virtue of the legal provisions currently in force, providing each citizen with equal opportunities. In this way, it deals with a contemporary concept which is aligned with the educational principles shared by developed countries. An essential correlate is that of the identification of differences between individuals and the importance of efficiently channelling resources in order to achieve personalisation [3,9,10]. This is important in pursuing the goal that no student drops out of the educational system without reaching at least a minimum standard of learning that will allow them to fully exercise their civil rights [10].

Attention to diversity is born out of a systemic focus, where educational institutions are considered to be responsible for the design of actions that bring about appropriate responses to the different contextual situations faced. This occurs in consideration of the relative legislative provisions in relation to the scope of the action [1,8,10].

### 1.1. Plans for Attention to Diversity

Attention to diversity constitutes an influential aspect of the educational centre’s quality system, in that it depicts its capacity to respond to the particular needs of the relevant educational community [10,11,12,13,14,15,16] and it is configured as one of the possible routes through which equality in democratic society can be achieved [8,17]. It establishes mechanisms through which training processes are adapted to the specific needs of each student, without segregating them from the other students [8,9,18]. Authors such as Azorín [14] and Rodríguez [19] indicate that the reduction of failure and dropout rates depends on the effectiveness of attention. These constitute two important challenges for the educational system, which will have a large repercussion on society in terms of their influence upon employability and quality of life.

The inclusive school, therefore, attends to all of their students regardless of whether or not they have Special Educational Needs (NEE) [13], individualising the training process and moving away from standardisation. Fitting with this, the objectives of education are allied with the needs of those receiving it, seeking to compose instruments that are diverse in nature, with many of these being designed ad-hoc, in order to present all students with the opportunity to receive equitable and viable learning [13].

Arnáiz and Guirao [20] have indicated that it is not only through inclusive education that a correct defence of human rights can be guaranteed in society, as this dimension also impacts upon key aspects such as equity and social justice, without which it is not possible to ensure full exercise of these rights or their adequate protection [21].

According to Hornby [21], inclusive education and special education are based on opposing principles and, as a result, they have led to the development of two contrasting models of attention to diversity, which become more distantly related as time goes on. Taking the viewpoint of Clark, Dyson and Millward [22], the first of these is currently considered to be more relevant as it is under this model that teaching and learning strategies are promoted to give all students within a shared environment access to the curriculum. Domínguez, López and Vázquez [23] indicate that following these two possible alternatives (inclusive education and special education), the concepts of equality and diversity can be confronted, advocating first of all for the former as an omnipresent issue in all social spheres. From this perspective, the challenge is constituted by offering attention to diversity in a way that ensures equality, achieving quality education through transversal actions which must also be flexible to the needs of the student body [23].

Despite the importance of attention to diversity for contemporary societies, its development has been more theoretical than practical and more intelligible than tangible, partly due to the barriers and obstacles that exist in the educational system, urging its evolution [24]. In this sense, it has been suggested that it is transcendental that the very educational centres put into action their own resources, within their own scope for investigation, in order to speed up the responses offered to students with NEE, adjusting their methodological strategies to the inclusive principles which must govern society. The Organic law for the improvement of educational quality (LOMCE) establishes attention to diversity as one of the central elements which must inform the way educational policies are programmed and the practices used. Thus, attention to diversity is set as one of the pillars of inclusive society [8]. The means through which attention to diversity is structured forms part of the Educational Project of the Centre. Amongst the possible measures, curricular and organisational adaptation are cited, amongst others [8,10].

At present, educational institutions are oriented towards quality, designing strategies based on this element through factors that induce a competitive improvement in their services. This circumstance has led to growth in the number of centres that count on quality management systems certified according to Pascual [25]. These are based on the true meaning of the concept of quality education that is linked to the promotion of equal opportunities. This requires the embedding mechanisms that operate at different levels (organisational, curricular, etc.) and can be measured in order to control and inform about the achievements made in relation to different dimensions. This has been certified by auditing companies [26]. According to Pfeffer [27], it must be considered that a quality education is equivalent to equality of opportunities. This equality cannot be afforded without an appropriate system that enables a deepening of the institutional channels which make it possible to guarantee it. As a result, these dimensions act as intertwined cogs which promote overall adjustments to suit the characteristics of the environments within which equality is promoted.

From this viewpoint, the present study has the goal of analysing the perceptions of Andalusian teachers who belong to educational centres within which a Management System for quality has been implanted. Their plans for attention to diversity, at both a general and academic level, will be examined.

### 1.2. Degree of Compliance with Strategies for Attention to Diversity

In accordance with Giné [1], questions in the present study were not posed from the perspective of any prism that education should be comprehensive and inclusive. However, it is crucial to provide evidence about the standpoint taken by each educational centre in relation to achievement of the objectives relating to attention to diversity, detecting potential gaps that may limit its effectiveness. In this regard, Giné [1] states that amongst the real difficulties of attention to diversity, the following are identified: lack of mechanisms to recognise the diversity of the student body, absence of connections between the different services and available resources and lack of links between the strategy for attention to diversity that operates in the centre and the actions implemented in other environments (such as within the family or community).

In this sense, it must be conceded that attention to diversity should be conceived from a point of transversality. This requires a participatory perspective from which integrated initiatives embrace the different levels of overlapping actions relating to each individual [1]. According to Escarbajal et al. [28], the educational system in Spain has not achieved its full potential, despite the fact that the economic endowment allocated to it is higher than the average of the Organisation for Economic Cooperation and Development (OCDE). According to Escarbajal et al. [28], the difficulties faced by an inclusive school trying to be effective are not derived from the lack of resources nor from the lack of rules that regulate teaching practice. Instead, they are generated by the way in which centres are managed, this being the arena where the most important challenges lie.

According to Gento [29], the policies of the centre should be conceived in order to achieve full development of all individuals and to contribute to the integration of each person in all potential spheres of participation. Taking this social approach, individuality must be understood through policies based on its latent characteristics. Fulfilment of attention to diversity is evidenced from a number of perspectives and dimensions, making equity tangible in all the required facets. However, difficulty is found at the level of the centre, since achieving full and equitable literacy of all students generating equivalent outputs, demands constant action and evaluation. These are areas in which Spanish educational centres have still not achieved enough, failing to efficiently delegate the full extent of their opportunities and resources [4,28,29].

## 2. Materials and Methods

The objective was to gather relevant, general and nuanced information about the aforementioned problem, making inferences without manipulating the independent variables in any way.

Selection of the surveying method for the present research sought to know the teachers’ perception with regards to the Improvement Strategies that target diversity in Compulsory Secondary Education. This is because there was a lack of information on what the main protagonists, the teachers, think in relation to this topic. To address this, we state here our agreement with Kerlinger [30], who states that “it is significant that even though hundreds of thousands of words are written and spoken about education and what people think about it, little reliable information is available on the topic” [30].

### 2.1. Participants

The population and study sample of the present research is constituted by a group of teachers who deliver teaching within the three Secondary Education Centres that have integrated Quality Management Systems in the Autonomous Community of Andalusia. Centres were selected for inclusion using a convenience sampling strategy. The selected centres were those which demonstrated willingness to participate in the research.

Taking as a basis the information provided by the directors of the Centres, the population was made up of 135 teachers. All of these identified teachers were invited to participate in the research, with 112 teachers providing data. This represents 82.96% of the overall population invited, of which 50 (44.8%) were male and 63 (55.2%) female.

Considering the teachers trained in Therapeutic Pedagogy, the average length of teaching experience was 13.6 years, relative to an average of 17.9 years of experience amongst those who did not possess this qualification. Consequently, this reveals that teachers who are not qualified in this area are in possession of greater teaching experience. Table 1 presents the distribution, frequency and percentage of teachers that constitute the sample.

### 2.2. Instrument

Just as has been presented, the instrument sought to obtain the perception of teaching staff relating to improvement strategies and attention to diversity. Thus, having reviewed the existing literature and checked that there was no instrument available that could adequately address our objectives, we proceeded to develop our own.

An ad hoc questionnaire for the evaluation of the quality of strategies addressing diversity was, therefore, the instrument used. This paper and pen tool was divided into the following dimensions: school environment, definition of the improvement strategy, planning of the improvement strategy, curricular design, human resources addressing diversity, equipment and resources. These are composed by 10, 8, 5, 6, 6 and 3 items, respectively. The instrument included a series of closed questions rated along a Likert-type response scale running from 1 to 5 (1 being not important and 5 being very important) which describes the degree of importance given by the teachers to the different dimensions. These dimensions must be considered when attending to diversity and improving quality within educational centres.

In order to assure content validity of the instrument, the opinion of a group of nine teachers was taken into consideration. This included teachers from the Faculty of Education Sciences working in the ambit of methodology, alongside secondary education teachers who were trained in Therapeutic Pedagogy and worked in the ambit of attention to diversity.

The number of experts was nine and a numeric rating scale was used which ranged between 1 and 4, with 1 being nothing and 4 being totally.

Thus, the highest score that could be obtained for both the dimensions and the items was 9 × 4 = 36. We established 9 × 3 = 27 as a cut point, considering 1 and 2 negative, and 3 and 4 positive. In this first part, six dimensions were validated: Environment of the educational centre, definition of the improvement strategy, planning of the improvement strategy, educational programs for the students, human resources addressing diversity, equipment and resources. First, we proceeded to validate the dimensions, followed by the items. As a result of validation, the name of the dimension “educational programs for the students” was modified and the title “curricular design” was instead used. Further, of the 57 items initially proposed, 53 items remained in the definitive version.

After carrying out content validity, we conducted a factorial analysis in order to establish construct validity of all of the dimensions. Following Rodríguez, Gil and García [31], this was carried out attending to the conditions of application, determination of commonalities, the extraction method and the rotation method.

For the study of internal consistency of the questionnaire, the reliability estimation coefficient, Cronbach alpha, was calculated. In Table 2, we present data for both the dimensions and the overall instrument. We highlight that the reliability coefficient value observed in all cases is sufficiently high, serving to validate the inferences and conclusions drawn in the present research.

### 2.3. Procedure

The process began through interviews with the directors of each one of the participating educational centres. Participants were explained all of the details of the research and they then verified their motivation and commitment to participate, making administration of the instrument possible.

A researcher from the study team travelled to each educational centre in order to administer the questionnaires to the teaching staff. The teachers who participated in the study did so voluntarily and confidentiality of the results was guaranteed. Once the completed instruments were collected, the coding, ordering and computer recording of the responses in the database was carried out for their subsequent statistical treatment in SPSS version 25.0 statistical package for Macintosh (IBM ® SPSS® Statistics 25) (Manufacturer name, City, State Abb, Country).

## 3. Results

Subsequently, the level of importance conferred by participating teachers to each of the dimensions asked about was analysed.

### 3.1. School Environment

As we can observe, the items that obtained a higher value on behalf of the teaching staff make reference to the sociocultural characteristics and needs of the immediate environment and relate to the importance of possible interactions with families and with tutors (Figure 1).

In contrast, those which obtained a lower rating described the possibilities for interaction with the institutions and associations found in the neighbourhood and municipality. Another low rating was found for the possibilities of professional integration of students with special educational needs. This indicates that these variables would not be considered of relevance when carrying out a plan for quality attention to diversity (Figure 2).

### 3.2. Definition of the Improvement Strategy

With regards to this dimension, the items which obtained more positive values are: “Degree of consensus on the school’s objectives for addressing diversity”, “available resources” and “physical and environmental infrastructures to meet the diversity of the centre” (Table 3).

In the rest of the items, there was not a large difference between the values reported. The items which stood out due to being given the lowest ratings by teaching staff were knowledge of the School Board about the needs of the student body, with 7.8%, and the consequences of not putting certain improvements into place, with 4.8%.

### 3.3. Planning of the Improvement Strategy

In this third dimension, the items rated of greatest importance by teaching staff for carrying out an Improvement Strategy for the aforementioned relevant characteristics are: “The strong and weak points of the educational centre must be determined in order to address diversity” and “the degree of agreement between members of the educational community regarding the actions that should take priority for the improvement of the quality of attention to diversity” (Figure 3).

In this case, no negative values stood out as they were almost indiscernible differences between each item (Figure 4).

### 3.4. Curricular Design

In Table 4, we can see that curricular design at the moment of carrying out an improvement strategy in an educational centre is a relevant aspect for teaching staff. As shown by the results, the majority of means are above a value of 4.

Appropriateness of the adaptations are considered by 99% of teachers to be an important mechanism for attending to the objectives of diversity. An identical percentage of teachers also rated the design of evaluations tailored to the actual curriculum being implemented in the classroom as important.

The item that refers to making changes in the organisation of teaching following an inclusion plan, and that relating to appropriateness of the curricular competence of each student with NEE to predetermined objectives and goals, are considered to be important aspects for 96.2% of participants. With respect to appropriateness of the teacher–student ratio in attention to the characteristics of the student body, this was reported to be of great relevance for 95.2% of respondents.

One of the questions to which a similar consensus was reached between the participating teaching staff was that relating to coordination and joint working of the department.

### 3.5. Human Resources for Attending to Diversity

With regards to the importance of counting on human resources for attending to diversity, very high results were found overall. When considering the professional training of teachers charged with assisting NEE children in regular classrooms, this factor was considered by 97.1% of teachers to have a hugely positive influence.

Likewise, for 99% of teachers, counting on voluntary teaching staff who contribute to the training of their colleagues with the aim of improving the quality of outcomes was one of the most valued indicators. Together with this, 100% of teachers require ongoing training.

### 3.6. Equipment and Resources

Teaching staff considered equipment and resources to be essential for implementing a strategy to address the quality of diversity (Table 5).

Aspects which stand out as being very important are the appropriateness and novelty of equipment and resources, the quantity and quality of specific equipment for addressing special educational needs (97.1%) and the adaptability, functionality and security of the facilities of the educational centres and inside its classrooms (98.1%).

Finally, the existing variables relating to the gender variable are presented in Table 6. We focused on the seven items that show significant differences between the opinions of males and females.

For the items “characteristics of the entire student body of the educational centre”, “possibilities for professional integration of students with special educational needs” and “the organisational structure in place to meet the objectives of the improvement strategy”, it can be highlighted that the means are higher in the female subgroup. In contrast, in the male group, responses to the questions regarding “limitations to achieving agreed objectives”, “a measurement approach must be developed” and “appropriateness of the teacher–student ratio to the characteristics of students with NEE in each classroom and educational centre”, presented a greater score.

## 4. Discussion

According to Muñoz and Espiñeira [11], evaluation of the quality of attention to diversity is contemplated from a consideration of the social standard as “ideal”, to which the standards of quality management strive towards. This establishes a similar range for all individuals from which they can identify homogenous progress expectations. Arnáiz and Azorín [32] similarly suggest that evaluation is the seed of improvement within processes, with use of this informative tool enabling the discovery of optimal routes in the educational process.

In this context and in line with the contributions made by Kondra and Hinings [33] and Gento [29], in the past—conceived from an anthropological perspective—difference was conceived as a deviation that could be redirected through intervention. However, in the present, the social dimension of education is advocated, from which it is considered that the system should welcome difference as a prime material. The link between attention to diversity and quality resides in the capacity of the centre to offer individual responses to each student without questioning. This aspect is also aligned with excellence.

The objective of the research consisted of better understanding the perception of teachers who practice their profession in centres within which a quality certification system relating to strategies for attention to diversity is in place. This included taking a general or idealist conception of the systems as well as considering those implemented in an educational ambit with the purpose of deepening understanding about their own characteristics. In this regard, it must be emphasised that to educate about diversity is nuanced and not about “straightforward” sermonising under the instruction of policies or rules. Instead, it is about being able to develop and comply with the generated measures [11]. As a consequence, the degree of compliance is as important as the existence of ad hoc plans designed by any organisation in order to address difference.

Attention to diversity constitutes an effort that requires aligning resources, policies and people, to a common philosophy from which the social interest of inclusive education and its fundamental character are emphasised in order to favour equality in society [9,29,34]. Due to the need to address the plurality of demands than can be identified in a particular scenario, full knowledge and recognition of the needs that may arise is needed, in addition to a firm commitment on behalf of all involved agents to strive towards achieving a quality educational system for all those within it [29].

For the teachers who participated in the study, it is crucial to provide opportunities for interaction with families and tutors, and to agree on the objectives of the educational centre for addressing diversity. Likewise, it is important to define a strategy for optimising resources which is improved by virtue of their availability in the immediate environment of the educational centre. Further, it is key to establish an appropriate teacher–student ratio in order to manage to develop individualised strategies and enact the tracking that each student needs. For this, it is necessary to coordinate the actions that all members of teaching staff will deploy. This aspect has also been highlighted by Escarbajal et al. [28]. In reality, coherence between the adaptations and their connection with the centre’s objective must be made clear in the various instruments of the centre (PEC, programming, etc.) and the way in which attention to diversity is reflected in the specified environment must be exhibited.

Teachers play an essential role in the development of strategies for attention to diversity [14] and as a result, they are the guarantors of an inclusive school. Recognising deficient aspects through their opinions provides a route through which the dimensions are revealed which can lead to an improvement in the development of these strategies.

It was revealed that teachers had to consolidate their knowledge and update it in order to be able to appropriately address the NEE of their students. This aspect has also been highlighted by Azorín [14]. In the same sense, Domínguez and Vázquez [35] have recognised the importance of familiarising teachers with the place of permanent training in adequate professional development, an aspect upon which, in agreement with the present results, there already exists adequate awareness amongst teachers. For Cejudo et al. [10], who focused their investigation exclusively on the preparation of teachers to address diversity, knowledge of teaching staff is deficient as a result of the lack of attention given to this dimension in the curriculum, despite it constituting one of the axes of education in the present day. In their research, a lack of knowledge is displayed in relation to both cultural and deficit manifestations of diversity.

## 5. Conclusions

The degree of compliance with plans for attention to diversity of the variables was considered to be of importance, showing the power of directing greater attention to them in educational centres in order to reach the consensus and coherence necessary between the actions designed and the needs of their recipients. Failure to do so will not enable achievement of the ideal for equality of opportunities, towards which the current educational system should orient.

In this respect, authors such as González et al. [36] have insisted on the need for greater training of teachers with the purpose of favouring optimisation of measures of attention. At the same time, this could be a route through which the quality of attention offered in the educational context is elevated.

In this context, diversity must be contemplated as an ever present need in educational centres, making it necessary to engage in continuous reflexion and evolution, in which the centres must constantly adapt in order to achieve educational quality and, ultimately, social justice in the educational environment [28].

## Figures and Tables

**Figure 1 ijerph-17-03840-f001:**
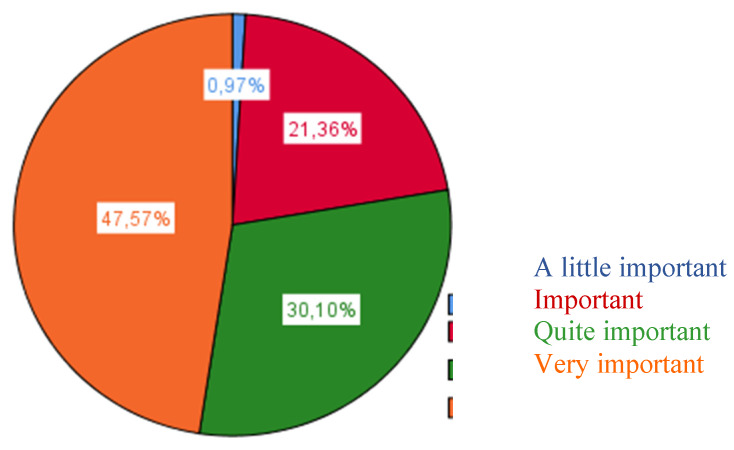
Sociocultural characteristics and needs of the immediate environment.

**Figure 2 ijerph-17-03840-f002:**
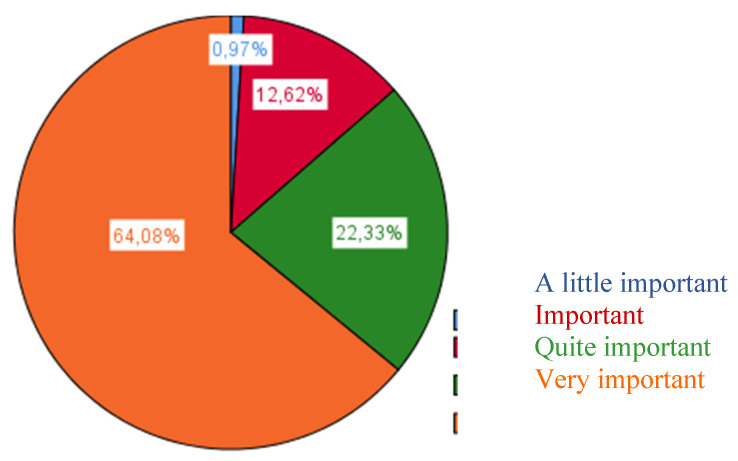
Potential for interaction with families or tutors.

**Figure 3 ijerph-17-03840-f003:**
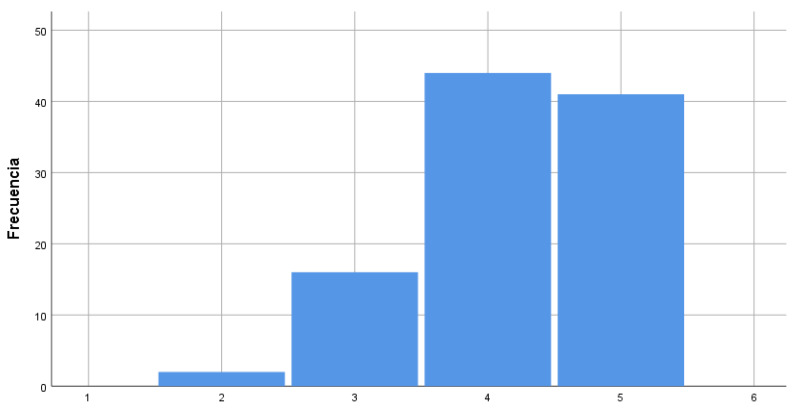
Strong and weak points of the educational centre for attending to diversity.

**Figure 4 ijerph-17-03840-f004:**
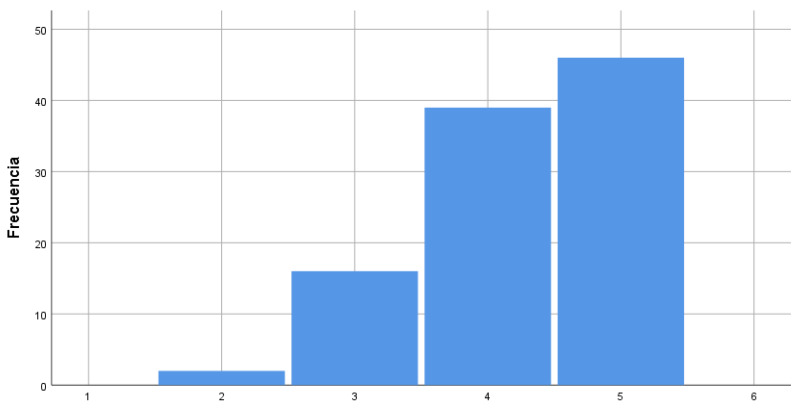
Degree of agreement about the actions required for improvement of the quality of attention to diversity.

**Table 1 ijerph-17-03840-t001:** Representation of teachers from the population in the sample.

Typology of the Centre	Total Teaching Staff of the Centre	Participating Teaching Staff	Participation Percentage
**Quality Centre A**	41	37	94.24%
**Quality Centre B**	48	45	93.75%
**Quality Centre C**	46	30	65.22%
**Total**	135	112	

**Table 2 ijerph-17-03840-t002:** Table of Cronbach alpha results.

	Alpha
Overall questionnaire	0.9569
School environment	0.8641
Definition of improvement strategy	0.8893
Planning of improvement strategy	0.8713
Curricular design	0.8947
Human resources addressing diversity	0.8050
Equipment and resources	0.8938

**Table 3 ijerph-17-03840-t003:** Definition of the improvement strategy.

	Degree of Consensus in the Objectives of the Centre for Attending to Diversity	Available Resources	Physical and Environmental Infrastructures to Meet the Diversity of the Centre
Not important	1.0%	1.0%	1.0%
A little important	1.9%	3.9%	2.9%
Important	18.4%	24.3%	22.3%
Quite important	45.6%	23.3%	34.0%
Very important	33.0%	47.6%	39.8%
Total	100.0%	100.0%	100.0%

**Table 4 ijerph-17-03840-t004:** Aspects of curricular design.

Item	Mean	Standard Deviation
Modification of the organisation of teaching following the inclusion plan.	3.93	0.867
Appropriateness of the curricular competence of each student with Special Educational Needs (NEE) to the predetermined objectives and goals.	4.09	0.931
Appropriateness of the teacher-student ratio to the characteristics of the students with NEE in each classroom and school.	4.37	0.889
Coordination and joint working of the entire department	4.44	0.730
Appropriateness of the adaptations of Education Plan (PEC) and Curriculum Project (PCC) in order to give effective responses to the objectives of the school relating to attention to diversity.	4.15	0.722
Design and use of evaluations based on the real curriculum received by students in the classrooms and school.	4.17	0.789

**Table 5 ijerph-17-03840-t005:** Equipment and resources.

	Appropriateness and Novelty of the Special Equipment and Resources for Addressing Diversity in Each Centre	Quantity and Quality of Specific Equipment Related with NEE	Adaptability, Functionality and Safety of All of the Centre’s Facilities, Especially in the Classrooms
Not at all important	1.0%	0%	0%
A little important	2.9%	2.9%	1.9%
Important	17.5%	16.5%	22.3%
Quite important	44.7%	44.7%	36.9%
Very important	34.0%	35.9%	38.8%
Total	100.0%	100.0%	100.0%

**Table 6 ijerph-17-03840-t006:** Differences in relation to the gender variable.

Items	Sex	Mean	Sig.
Characteristics of all of the students at the educational centre.	Male	3.77	0.001
	Female	4.39	
Potential for professional integration of students with special educational needs.	Male	3.34	0.000
	Female	4.05	
Collaboration through cooperative programs with other educational and social institutions.	Male	3.43	0.012
	Female	3.88	
The organisational structure in place to respond to the objectives of the improvement strategy.	Male	3.58	0.001
	Female	4.14	
Limitations to achieving agreed objectives.	Male	3.92	0.023
	Female	3.58	
An approach for measurement must be developed.	Male	4.28	0.000
	Female	3.81	
Appropriateness of the teacher–student ratio to the characteristics of students with NEE in each classroom and educational centre.	Male	4.32	0.005
	Female	3.83	

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
