# Peer review of "Strategies for Attention to Diversity: Perceptions of Secondary School Teaching Staff"

_ijerph, 2020, doi:10.3390/ijerph17113840_

Round 1
Reviewer 1 Report
For the following reasons I recommend to reject this paper:
(1) The overarching theme of the paper is diversity among students in school education. In my opinion, this theme does not match the aims and scope of the International Journal of Environmental Research and Public Health.
(2) It remains unclear to me which construct the authors examine in their study. Do they examine teachers' perceptions (line 105), their opinions (line 138), their reflections (line 140), their thinking (line 142), or their attitudes (line 165) towards the question of how to deal with diversity in school education? In my opinion, the authors should have clarified this question in their paper.
(3) The description of the questionnaire used in this study is very fragmentary. For example, it remains unclear to me which ideas underlay the development of the questionnaire, how the validation took place, and what findings the validation provided. Especially since there is a lack of questionnaires for surveying teachers' opinions and attitudes towards the questions of how to address students' diversity (line 165-168), I would be very eager to read a paper about the development and validation of such a questionnaire.
(4) Especially since the description of the questionnaire left so many questions open for me, I do not consider myself to be in a position to judge the authors' results, discussion, and conclusion.
Reviewer 2 Report
Abstract - Revision for clarity would be helpful. As currently written, it may be difficult for readers to comprehend the overall purpose and results of the study.
The last two sentences in the abstract are of some concern as written. Attention to diversity could be one of the most significant challenges facing education. It is vital teachers maintain respect and appreciation for differences, but not all educators hold those beliefs; thus, opinions of those educators would not be a successful route for achieving equity and efficacy.
Introduction - Wording is awkward in many places. Examples include lines 40-44 and 47-50. Word choice changes would make the content more coherent.
Delete numerous introductory clauses for improved clarity. One example is line 61 - eliminate "As a consequence." Intensive revision for word choice and clarity would significantly improve the introduction. Lines 60-65 are not clear nor concise.
The authors' wording is a bit awkward and lacks fluency and is shadowing the importance of the content.
Possible revisions might be eliminating "In this respect" in 75.
79-81 is a one-sentence paragraph that needs revision.
118-120 reword
Materials and Methods - Reads that study was to learn what teachers thought about the school's improvement strategy for attention to diversity; however, stated in other areas is the following:
15-17 - Teachers' perceptions of the level of importance assigned to variables in plans will influence the degree of compliance (Was the degree of compliance measured?)
22-25 - Changing teacher opinions is necessary to improve methods for quality and inclusion (How is that possible, especially if the teacher has inadequate respect and appreciation for differences?)
124-125 - Management of centers (What management concerns?)
126-134 - Policies of the center (What specific policies?)
133-134 - Centers failing to use opportunities (What opportunities?) and resources (What resources?) fully
Questions from a reader:
Did the centers clearly articulate to teachers their proposed responses to challenges related to diversity?
Were teachers trained?
Were teachers part of the review, design, implementation, and then the monitoring of progress and improvement strategies?
Are the teachers researchers of attention to diversity?
144-147 Is the little reliable information available related to education or about attention to diversity?
160-161 Does not having the training in Therapeutic Pedagogy mean the teachers are not qualified or that teachers without training in Therapeutic Pedagogy possess greater years of teaching experience than those trained in Therapeutic Pedagogy?
163 Correction to Table 1
Instrument - The desired instrument was one to measure opinion and attitude related to improvement strategies AND attention to diversity. Were opinions and attitudes differentiated?
169-170 - A questionnaire was designed to evaluate the quality of the strategies addressing diversity. How does this coincide with measuring the opinion and attitude of the teachers for the strategies as well as the opinion and attitude of the teachers regarding attention to diversity?
175-177 - Measuring the degree of importance the teachers gave to the dimensions of the survey? (But also stating in 176-177 that these dimensions must be considered, so if teachers did not assign importance to what must be considered what does this reveal?)
Perhaps revise this section for improved clarity.
200 Corrections to Table 2
Procedures - Again, rewording would be helpful. Example 202-203: The process began with an interview with the director from each participating educational center.
Results - states the level of importance conferred by teachers to the dimensions was analyzed. Perhaps revise the variety of proposed measurements described in the methods section.
Consider using a different method to display data in Figures 1 and 2.
232-233 - delete "This being said"
Discussions and Conclusion
296-"with use of this informative tool" - what tool? Consider revision for clarity.
Wording the findings with clear justification from the results of the study would improve this section.
Reviewer 3 Report
I find the topic of this study important. The article is well-structured and easy to read. I would just suggest labelling all figure legends in English. In addition, the content of the article could be improved by additional qualitative research in form of interviews with a selected secondary school teachers in order to elicit more in-depth information about the improvement strategies aimed at diversity.
Round 2
Reviewer 1 Report
The quality of the paper has noticeably improved. However, it still requires revision before publication. Although the authors write that they investigate the perceptions of teachers, I still have almost no idea what construct the authors are investigating. In the educational sciences, teachers' perceptions are regarded as an "messy" construct (e. g. Pajares, 1992) for which there are a multitude of (sometimes contradictory) conceptualisation. Therefore, in order to provide clarity for the reader and to prevent misunderstandings, I think it is imperative that the authors explicitly and comprehensively describe which understanding of the teacher's perception underlies their study.
Pajares, M. F. (1992). Teachers’ Beliefs and Educational Research: Cleaning Up a Messy Construct. Review of Educational Research, 62(3), 307–332.
Author Response
The amendment requested by the reviewer within the article, in lines 38-47, has been responded to.